# Deep Learning-Based Algal Bloom Identification Method from Remote Sensing Images—Take China's Chaohu Lake as an Example

**Shengyuan Zhu [1], Yinglei Wu [1] and Xiaoshuang Ma [2],***

1    China JIKAN Research Institute of Engineering Investigations and Design, Co., Ltd., Xi'an 710000, China
2    School of Resources and Environmental Engineering, Anhui University, Hefei 230601, China
*    Correspondence: mxs.88@whu.edu.cn

**Abstract:** Rapid and accurate monitoring of algal blooms using remote sensing techniques is an effective means for the prevention and control of algal blooms. Traditional methods often have difficulty achieving the balance between interpretative accuracy and efficiency. The advantages of a deep learning method bring new possibilities to the rapid and precise identification of algal blooms using images. In this paper, taking Chaohu Lake as the study area, a dual U-Net model (including a U-Net network for spring and winter and a U-Net network for summer and autumn) is proposed for the identification of algal blooms using remote sensing images according to the different traits of the algae in different seasons. First, the spectral reflection characteristics of the algae in Chaohu Lake in different seasons are analyzed, and sufficient samples are selected for the training of the proposed model. Then, by adding an attention gate architecture to the classical U-Net framework, which can enhance the capability of the network on feature extraction, the dual U-Net model is constructed and trained for the identification of algal blooms in different seasons. Finally, the identification results are obtained by inputting remote sensing data into the model. The experimental results show that the interpretation accuracy of the proposed deep learning model is higher than 90% in most cases with the fastest processing time being less than 10 s, which achieves much better performance than the traditional supervised classification method and also outperforms the single U-Net model using data of whole year as the training samples. Furthermore, the profiles of algal blooms are well-captured.

**Keywords:** algal blooms; water pollution; remote sensing; object identification; deep learning

## 1. Introduction

In addition to being a valuable source of wealth for human beings, lakes serve the functions of regulating river runoff, alleviating drought and flood disasters, and providing biological habitats [1,2]. Rapid economic development and population growth have introduced some serious environmental problems, including the eutrophication of lakes, which seriously endangers their ecological–environmental service functions [3]. One of the direct consequences of water eutrophication is the outbreak of algal blooms. High temperatures and high concentrations of nitrogen and phosphorus are two of the main inducements of algal blooms outbreaks. On the one hand, the algaes in aquatic environments play the role of key organisms, as they can contribute to ecological restoration and feed several other trophic levels, and some kinds of algae can even be processed into food with nutritional value for human consumption [4]. However, on the other hand, the enrichment of algae can also cause environmental problems. Algal blooms not only decrease water quality and damage ecosystems, but they also pose a serious ecological risk through direct environmental pollution [5]. Therefore, rapid and accurate monitoring of algal blooms is of great significance for the prevention and control of algal blooms.

The manual or automatic site sampling and monitoring method is a routine and intuitive method for water eutrophication monitoring [6]. Using physical and chemical

analyses of water samples, the traits of algae can be directly and accurately obtained. However, it is difficult for this traditional sampling and monitoring method to grasp the dynamic spatial distribution of algal blooms over a large scale. Furthermore, these approaches are generally expensive and labor-intensive [7,8]. As a good supplement, the remote sensing technique has the advantage of being more easily applicable on a large scale. By using remote sensing images, the water blooms in the whole area of a lake can be dynamically monitored with much less labor cost [9–12].

A number of methods have been presented in the last two decades to identify algal blooms in lakes using remote sensing images, mainly including the Normalized Difference Vegetation Index (NDVI) method [13], the Floating Algal Index (FAI) method [14], the traditional Support Vector Machine (SVM) method [15], and the Back Propagation (BP) neural network method [16]. Generally speaking, these traditional algal bloom identification methods can be categorized into two main classes [17], namely, the index threshold method and the supervised classification method. The index threshold method normally derives an index based on the principle that the chlorophyll in algae has a strong reflection on the near-infrared band spectrum but a weak reflection on the red band spectrum [18]. By setting a certain classification threshold, the task of automatically distinguishing algal blooms from water can be accomplished. Two typical indexes to identify the algal blooms are the NDVI [13] and the FAI [14]. The basic principle of the supervised classification method is to use known samples to constantly train and optimize feature parameters, and a discriminant function is then constructed as the judgment rule to realize the image classification task. Based on multi-source satellite images, Qi et al. [15] used the traditional SVM method to monitor water blooms on lakes. Zhang et al. [17] identified algal blooms in Taihu Lake through SVM and other methods, and then analyzed the impact factors and change trends of algal blooms based on meteorological and hydrological data. Qi et al. [16] proposed to identify water blooms based on a BP neural network method using sentinel-1 radar remote sensing images, which can effectively distinguish water blooms from water.

The challenges of algal bloom identification and the defects of existing fall under the following two main aspects: on the one hand, with the change in bloom concentration or algae type, the spectral reflection characteristics of blooms change, and the conventional algorithm has difficulty fully exploring the spectral characteristics of bloom diversity, limiting the applicability of the algorithm; on the other hand, water blooms show obvious aggregation effect in space, while conventional methods often ignore the spatial structure characteristics of blooms. In the last few years, deep learning theory has attracted wide attention in many fields [19]. In the field of remote sensing image interpretation, deep learning methods have also been widely studied and have shown some notable advantages compared to traditional methods [20–22]. With the help of a large number of labeled training samples, deep learning can adaptively mine the spectral and spatial structure information of remote sensing images, achieving the purpose of finding the typical features of targets of interest and accurately identifying them [23,24]. Additionally, once the deep learning model is well-trained, it can quickly finish the task of image interpretation, even for images of large sizes.

Clearly, deep learning theory provides the possibility for the rapid and accurate monitoring of algal blooms [25]. Taking Chaohu Lake as the study area, this paper focuses on the topic of algal bloom identification based on the deep learning method. To ensure the robustness of the proposed model, the spectral reflection characteristics of the algae in Chaohu Lake are analyzed; then, based on the findings about the traits of the algae in Chaohu Lake, a dual U-Net model (including a U-Net network for spring and winter and a U-Net network for summer and autumn) is employed, wherein the Position-Channel Attention (PCA) architecture is embedded to enhance the capability of the network on feature extraction.

The main contributions of this work are summarized as follows. The spectral reflection characteristics of the algae in Chaohu Lake in different seasons are analyzed, and different traits of the algae in different half-year time periods are found. Based on the above spectral

reflection characteristics of the algae and using the sufficient training samples, a dual deep learning-based model is constructed to monitor the algae in the half year spanning summer and autumn and the other half year spanning spring and winter, separately. Lastly, a Position-Channel Attention architecture is utilized to help the network extract the features of target.

The rest of this paper is organized as follows. Section 2 introduces the basic information about the study area and the remote sensing dataset used for algal bloom monitoring, and the spectral reflection traits of the algae in Chaohu Lake are analyzed. In Section 3, the Dual Position-Channel Attention U-Net (DPCAU-Net) model for algal bloom monitoring is proposed. Then, the experimental results are described in Section 4. Finally, we draw conclusions in Section 5.

## 2. Study Area and Analyses of Remote Sensing Data

### 2.1. Study Area and Remote Sensing Dataset

Chaohu Lake is the fifth-largest freshwater lake in China, which is located in the central part of Anhui Province, between latitudes 31°25′ and 31°42′ N and longitudes 117°17′ and 117°50′ E (Figure 1). The average water depth of Chaohu Lake is 3.10 m, the maximum water depth is 6.78 m, and the water area is about 780 km$^2$. The water storage of Chaohu Lake during the wet season is 32.3108 m$^3$, and 17.2108 m$^3$ during the dry season. The urban land area of the Chaohu Lake Basin increased by about 2 times in the last 30 years [26–28]. With population growth, urban expansion, and rapid development of industry and agriculture, a large amount of wastewater is discharged into Chaohu Lake, resulting in the intensified eutrophication of Chaohu Lake and the frequent outbreaks of algal blooms.

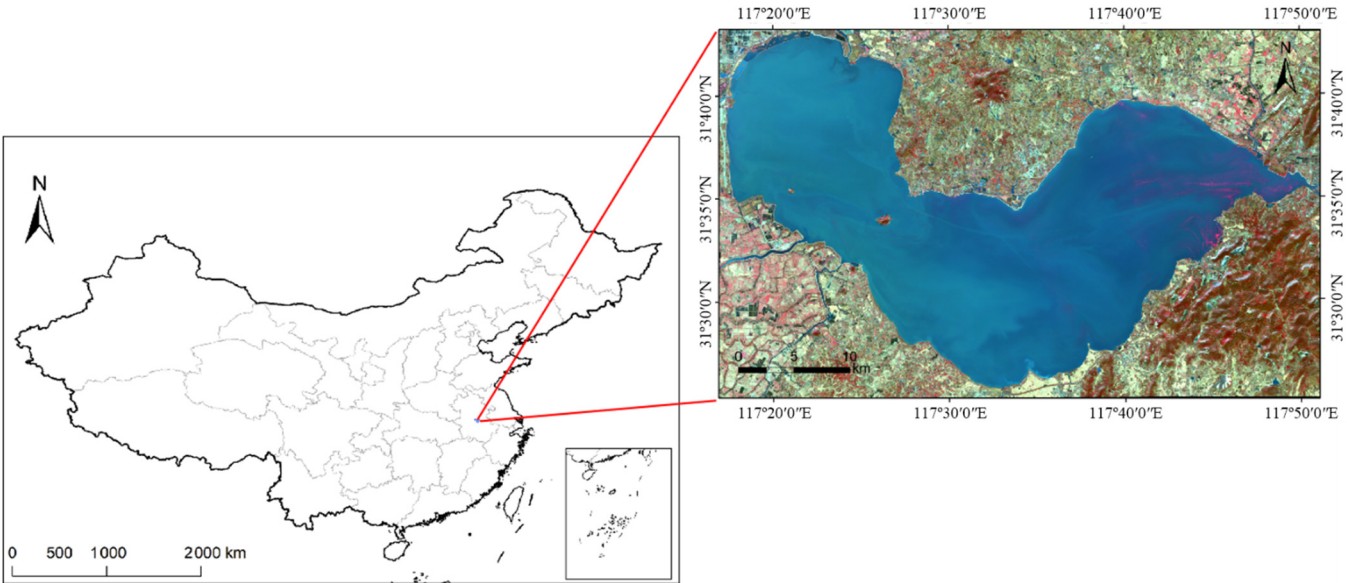

**Figure 1.** Map of the study area and the corresponding Sentinel-2 remote sensing image.

In this study, we used images acquired by the Sentinel-2 satellites (Sentinel-2A satellite and Sentinel-2B satellite) for the monitoring of algal blooms. The Sentinel-2 dataset can be freely downloaded from the European Space Agency (ESA) data distribution website (https://scihub.copernicus.eu/ (accessed on 10 December 2022)). The two Sentinel-2 satellites enable a short revisiting period (5 days) to observe the earth, and they can also provide multi-spectral images with high spatial resolution, which provides favorable conditions for users for monitoring the status of water blooms.

To train robust deep learning models, sufficient samples are necessary. In this study, 47 Sentinel-2 images from 2016 to 2019 in which algal blooms occurred were selected as the

dataset to select the training samples of the proposed model. The imaging periods of the 47 Sentinel-2 images used in this study are listed in Table 1.

**Table 1.** Imaging periods of the images used for training.

|       | Spring                | Summer   | Autumn   | Winter                |
|-------|-----------------------|----------|----------|-----------------------|
| 2016  | No cloud-free images  | 2 images | 1 image  | No cloud-free images  |
| 2017  | No cloud-free images  | 1 image  | 3 images | 3 images              |
| 2018  | 4 images              | 5 images | 6 images | 2 images              |
| 2019  | 5 images              | 2 images | 8 images | 5 images              |

A total of 400 samples with a patch size of 256 × 256 pixels each were selected from the above dataset, wherein the ratio of the algal bloom area to the water area was about 3:1. Before the model training and the final algal bloom monitoring processes, some preprocessing steps were taken to improve the quality of the remote sensing images, mainly including radiometric calibration and atmospheric correction, which were accomplished using ENVI 5.3 and SNAP 6.0 software.

*2.2. Analyses of the Spectral Reflection Traits of Algal Blooms in Chaohu Lake*

According to some researchers [29,30], the types of algae in Chaohu Lake show typically seasonal change trends, with microcystis outbreaks mainly occurring in summer and autumn, and dolichospermum outbreaks mainly occurring in spring and winter. At per-unit chlorophyll a concentrations, the reflection spectral power of microcystis is higher than that of dolichospermum [31]. Furthermore, the community size and cell diameter of algae also show seasonal change trends, being large mainly in spring and winter and small mainly in summer and autumn [32] Specifically, the chlorophyll a concentration per unit volume of water area in summer and autumn is relatively higher than that in spring and winter. To summarize, the spectral reflection traits of algal blooms is, in theory, much different between the half year spanning summer and autumn than the other half year spanning spring and winter.

To inspect the above issue, we selected some algal bloom samples from the remote sensing images and plotted the spectral reflection curves of the algal blooms that experienced an outbreak in different seasons. As can be observed in Figure 2, in all cases, the reflection of light in the red light band (with a wavelength of about 0.5 μm) and blue light band (with a wavelength of about 0.65 μm) are very low and the reflection of light in the green light band (with a wavelength of about 0.55 μm) is relatively high, due to the existence of chlorophyll in the algae. Furthermore, a strong reflection peak is exhibited in the near-infrared band (with a wavelength of about 0.8 μm), due to the special cellular structures of algae. However, we can also notice some differences. Generally speaking, the reflection curves of the water blooms in spring and winter are close to each other, while the curves in summer and autumn are much closer to each other and differ from those in spring and winter to some degree, especially as regards their much greater reflection in the infrared bands.

To further support the basic conclusion that, in Chaohu Lake, the spectral reflection traits of algal blooms in the half year spanning summer and autumn is different from the other half year spanning spring and winter, the NDVI values [13] and FAI values [14] of the samples are listed in Table 2. Clearly, the values for the algal blooms in summer and autumn are close to each other, while those in spring and winter are even closer to each other.

**Table 2.** NDVI and FAI values of the algal bloom outbreaks in different seasons.

|      | Spring | Summer | Autumn | Winter |
|------|--------|--------|--------|--------|
| NDVI | 0.3026 | 0.4162 | 0.4507 | 0.2678 |
| FAI  | 0.0765 | 0.1323 | 0.1376 | 0.0560 |

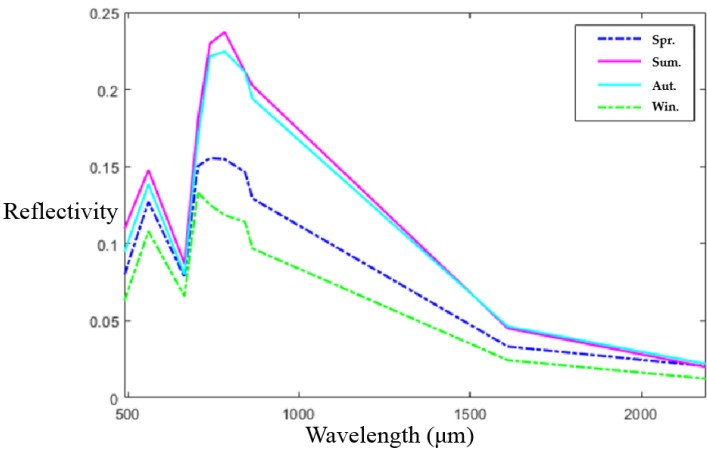

**Figure 2.** Spectral reflection curves of algal bloom outbreaks in different seasons.

## 3. Method

### 3.1. Basic Architecture of Deep Learning-Based Algal Bloom Monitoring Model

As mentioned, the challenges of bloom extraction and the defects in existing methods mainly include: On the one hand, with the change in bloom concentration or algae type, the spectral reflection characteristics of blooms can change, and the conventional algorithm has difficulty fully exploring the complex spectral characteristics of blooms, thus limiting the applicability of the algorithm; on the other hand, water blooms show obvious aggregation effect in space, while conventional methods often ignore the exploitation of spatial structure characteristics. Deep learning with a large number of marked training samples can adaptively mine the spectral and spatial information in images so as to accurately identify the targets. Among the deep learning networks, the Fully Convolutional Network (FCN) [33] is one of most popular semantic segmentation networks. By improving the previous convolutional neural network full connection layer and outputting images with context information, FCN can achieve good interpretation results. The U-Net model [34] is a typical FCN, which adds many jump links to the network to use the downsampled feature information during the upsampling reduction process, thus effectively retaining the structural texture information of the original image. Taking the U-Net model as the basic framework and considering the spectral reflection traits of algal blooms in Chaohu Lake, this paper presents a Dual Position-Channel Attention U-Net (DPCAU-Net) model for algal bloom monitoring in Chaohu Lake.

Normally, deep learning-based target identification methods include two steps, namely, the training step and the identification step. By inputting the feature vector $\gamma$ of the training samples, the purpose of training a U-Net model for algal bloom identification is to find the parameter set $\theta$ of network $f_\theta(\cdot)$, which ensures that the identification results $f_\theta(\gamma(i))$ and the real sample labels $a(i)$ are as similar as possible, as shown here:

$$\hat{\theta} = \text{argmin}_\theta \sum_i^M |f_\theta(\gamma(i)) - a(i)|, \tag{1}$$

where $M$ is the number of training samples. Once the optimal parameter set $\hat{\theta}$ is obtained, or in other words, once the model is well-trained, then given any pixel in the remote sensing image with feature vector $\gamma\prime$, the algal bloom identification result can be deduced by:

$$F = f_{\hat{\theta}}(\gamma\prime). \tag{2}$$

Taking the task of identifying algal blooms in a certain half-year (summer and autumn, or spring and winter) as an example, the basic architecture of the proposed model is shown in Figure 3. The model consists of two main parts, namely subsampling and upsampling. Downsampling is used to extract the image features, and upsampling is used to restore the segmentation results to the same resolution as the original image.

More specifically, the network includes dual convolutional modules, maximum pooling layers, and deconvolution layers, where the dual convolutional module consists of two Convolutional layer-Batch Normalization layer-Swish activation function layer (Conv-BN-Swish) modules. Furthermore, to enhance the capability of the network regarding feature extraction, a Position and Channel (PC) attention module was added.

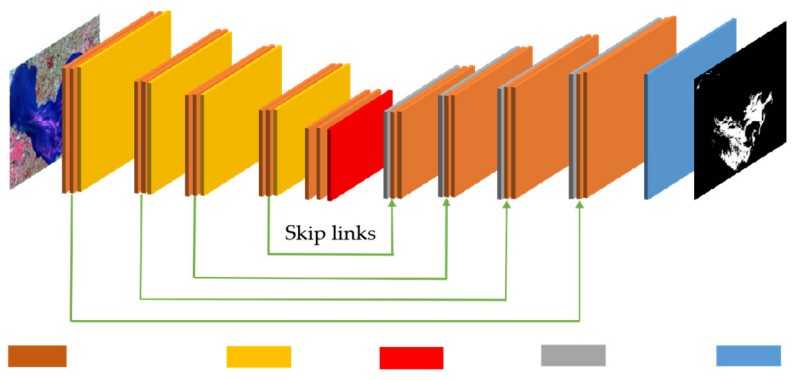

**Figure 3.** The basic architecture of the proposed DPCAU-Net model for algal bloom monitoring.

### 3.2. PC Attention Mechanism

In recent years, the attention mechanism [35] has been widely used in various deep learning tasks, such as natural language learning and image recognition, and is one of the important means to improve the learning ability of convolutional neural networks. The attention mechanism in deep learning is similar to the human visual attention mechanism. By using minimal attention resources, the network can pay more attention to the main features of the targets, thus improving the efficiency and accuracy of feature information extraction [36].

The position attention mechanism and channel attention mechanism are two typical attention mechanisms. The position attention mechanism focuses on the degree of spatial dependence between any two positions on the feature map. For the focused feature, the position is feature-weighted and updated, and the weight is the feature similarity between the two positions. Similarly, the channel attention mechanism focuses on the degree of spatial dependence between any two channels on the feature map. For the focused features, the channel is feature-weighted and updated, and the weight is the feature similarity between the two channels. In this study, to take advantage of the strengths of both the position attention mechanism and channel attention mechanism, we utilized the PC attention mechanism to enhance the capability of the network in extracting the texture feature information of algal blooms in remote sensing images. For the focused feature, the position is feature-weighted and updated, and the weight is the product of the feature similarity between two positions and the feature similarity between two channels.

In the training stage, the Adam optimization algorithm [34] was employed to adaptively update the network weights and biases, with the parameters setting as: $\beta_1 = 0$, $\beta_2 = 0.99$. The learning rate $\alpha$ was initialized as 0.001 for the network, and after every 40 epochs (the total training epoch limit was set to 200), $\alpha$ was multiplied by a decaying factor of 0.1 to decrease the searching range of the parameters.

## 4. Results and Discussion

### 4.1. Quantitative Assessment Indices

To quantitatively assess the performances of the different algal bloom identification methods, the widely used F1-Score index [11] was employed in the experimental part.

The F1-Score index is a combination of the Precision index and the Recall index. Precision represents the proportion of the correctly identified algal bloom pixels to the total number of the pixels that are identified as algal blooms by the classification method. Precision is calculated by:

$$\text{Precision} = \frac{TP}{TP + FP}, \tag{3}$$

where *TP* and *FP* denote the number of true positive and false positive samples, respectively. Recall represents the proportion of the correctly identified algal bloom pixels to the total number of the actual algal bloom pixels:

$$\text{Recall} = \frac{TP}{TP + FN} \tag{4}$$

where *FN* denotes the number of false negative samples.

The F1-Score is then given by:

$$\text{F1} - \text{Score} = \frac{2\text{Precision} \cdot \text{Recall}}{\text{Precision} + \text{Recall}}. \tag{5}$$

Normally, F1-Score is more objective than Precision and Recall for the assessment of image interpretation methods, since it is the harmonic average of the above two indices.

### 4.2. Algal Bloom Identification Results

In this part, the algal bloom identification results of the proposed DPCAU-Net model are reported. First, we used some samples from the images from 2016 to 2019 to train the DPCAU-Net, as described in Section 2.1, and we used the well-trained model to identify algal blooms in the images from 2016 to 2019. Then, to validate the good generalization capability of the DPCAU-Net in identifying algal blooms in different years, we directly employed the model trained using the samples from 2016 to 2019 to identify the algal blooms in the images from 2020. To illustrate the superiority of the proposed DPCAU-Net model, the algal bloom identification results by the Support Vector Machine (SVM) supervised classification method [15] and the Back Propagation Neural Network (BPNN) method [16] are also reported in this paper. Additionally, to further validate the necessity of employing a dual PCU-Net model (including a PCU-Net network for spring and winter and a PCU-Net network for summer and autumn), we also employed a Single PCU-Net (SPCU_Net) model to identify the algal bloom outbreaks in the different seasons.

In Table 3, the quantitative assessment results of the different methods for the spring and winter validation datasets and the summer and autumn validation datasets from 2016 to 2019 are listed. The image interpretation time of each method is also listed. To better illustrate the superiority of the proposed method, four remote sensing images from different seasons in 2019 and the corresponding interpretation maps by the different methods are displayed in Figure 4, wherein the remote sensing images are false color images, formed with reflectivity of the near-infrared band (red), red light band (green), and green light band (blue).

**Table 3.** Evaluation of algal bloom identification results in Chaohu Lake from 2016 to 2019.

| Method | Spring and Winter | Summer and Autumn | Interpretation Time |
|---|---|---|---|
| SVM | 77.82% | 79.47% | 41.4 s |
| BPNN | 79.09% | 89.88% | 62.1 s |
| SPCU_Net | 80.54% | 91.78% | 6.5 s |
| DPCU-Net | 91.89% | 97.31% | 7.2 s |

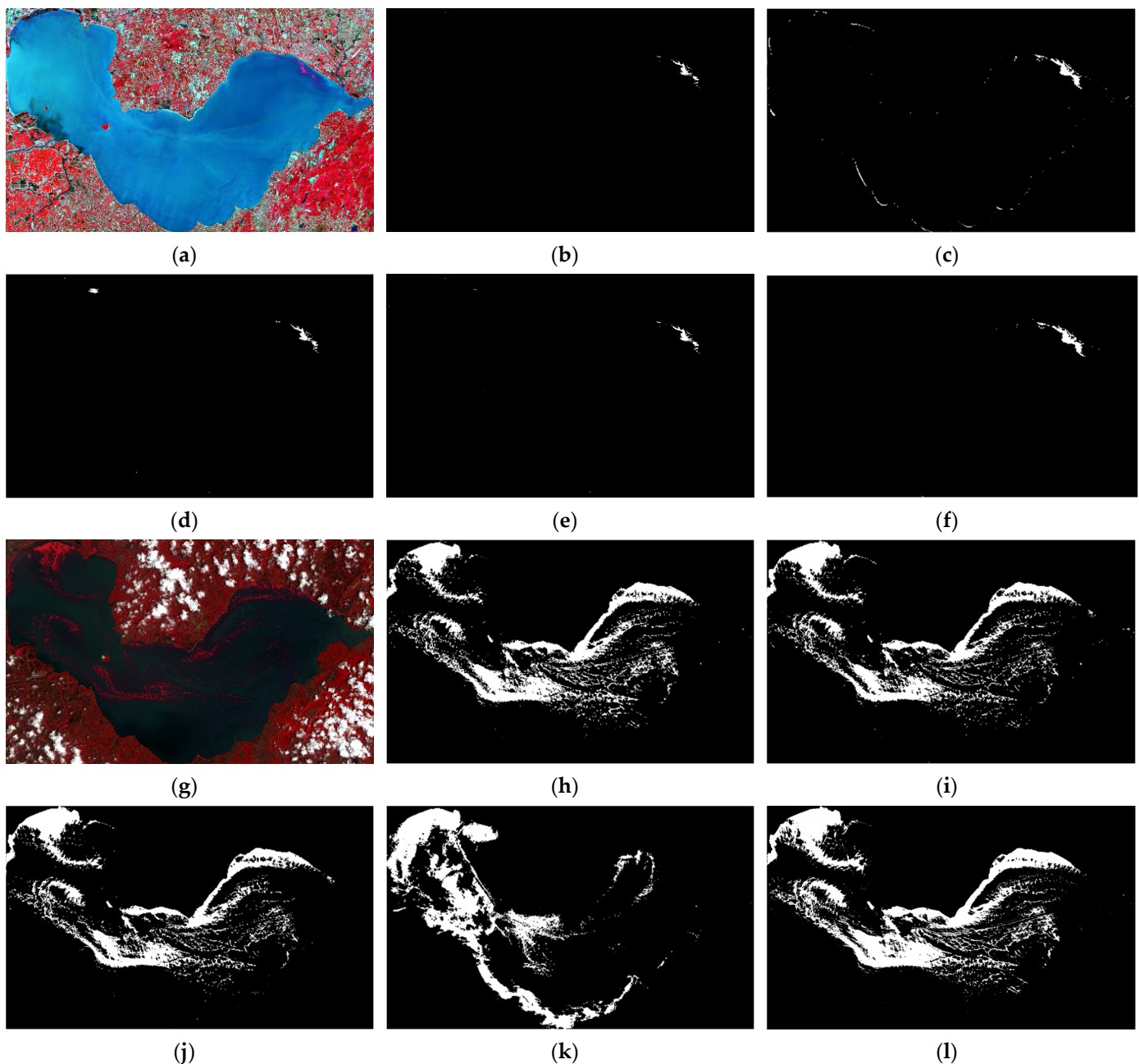

**Figure 4.** *Cont.*

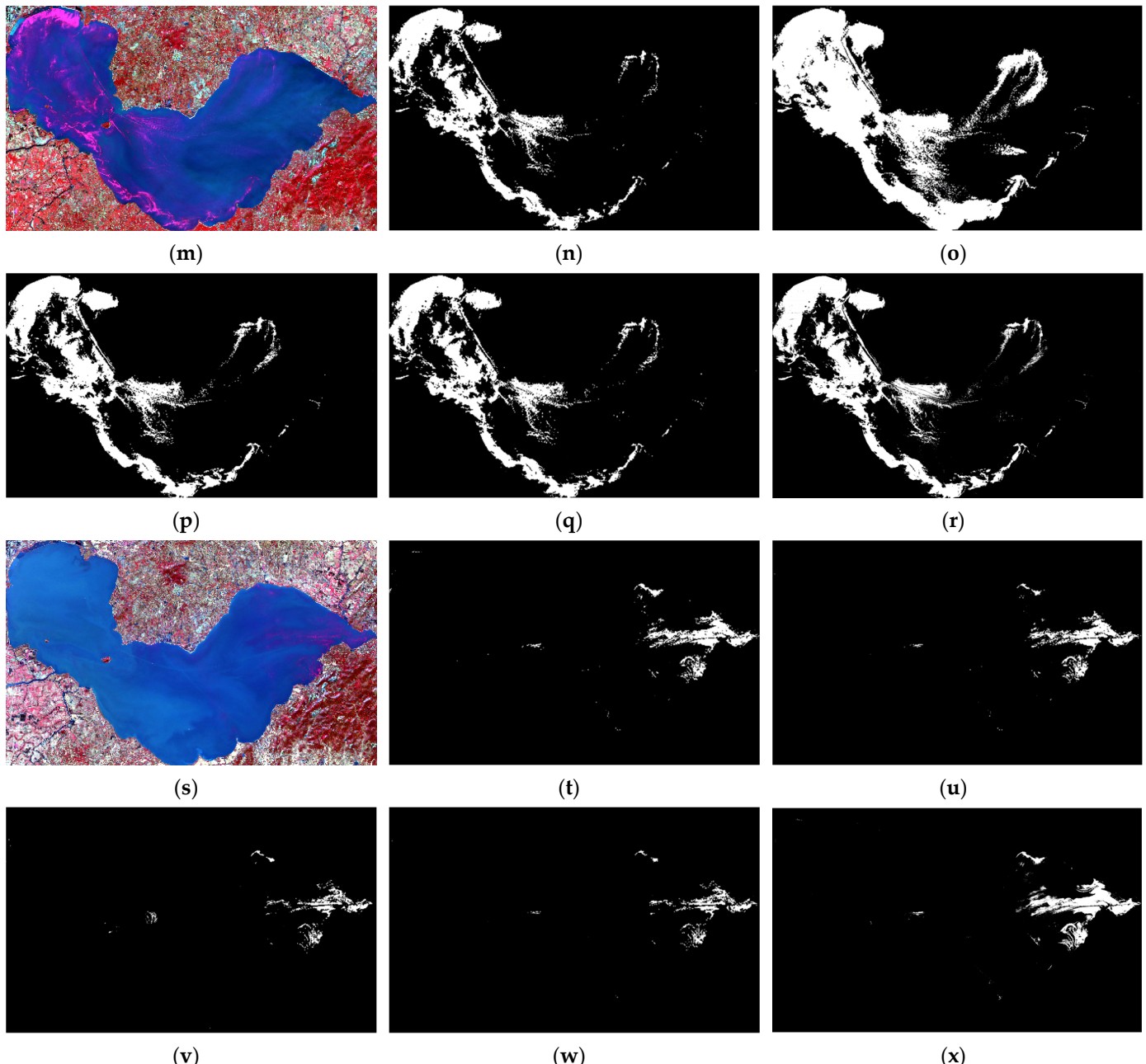

**Figure 4.** Algal bloom identification results for the images from 2019: (**a**) a remote sensing image of Chaohu Lake in spring; (**b**) a labeled image of (**a**); (**c–f**) the identification results by SVM, BPNN, SPCU_Net, and DPCU_Net, respectively; (**g**) a remote sensing image in summer; (**h**) a labeled image of (**g**); (**i–l**) the identification results using different methods; (**m**) a remote sensing image in autumn; (**n**) a labeled image of (**m**); (**o–r**) the identification results using different methods; (**s**) a remote sensing image in winter; (**t**) a labeled image of (**s**); (**u–x**) the identification results using different methods.

Clearly, from Table 3, we can see that the proposed DPCU-Net method performs best among all the methods, especially performing much better than the SVM method. This can also be supported by the observations in Figure 4. For the image in each season, the algal bloom identification map using the proposed method is highly in line with the real distribution state of algal blooms, or in other words, it is highly in line with the label image. The profiles of the algal blooms are well-retained using the proposed method. However, for the other three methods, the identification results are not always satisfactory. For example, for the spring case, some pixels near the boundary of Chaohu Lake are

misclassified as water blooms by the SVM method and the BPNN method; for the summer case, the identification performance of the SPCU_Net method for the middle areas of the lake is poor; for the autumn case, the identification performance of the SVM method for the western part of the lake is poor; and for the winter case, some pixels in the center of Chaohu Lake are misclassified as water blooms by the BPNN method. Furthermore, the SPCU_Net method and DPCU_Net method are also much more computationally efficient than the SVM and the BPNN methods, and the computational efficiency of BPNN is lowest among all the methods. In fact, once the model is well-trained, high computational efficiency is the common trait of the deep learning-based methods with regard to those traditional machine learning methods.

We also notice an interesting phenomenon in Table 3: the assessment values of the SVM method for the two validation sets in different half-year time periods are relatively comparable; however, the assessment values of each of the two deep learning-based methods are more varied, and the values in the summer and autumn validation set are much higher than that in the spring and winter validation set. This is because the deep learning-based methods are big-data driven and their good performances highly depend on a sufficient number of samples. In spring and winter, the occurrence frequency of algal bloom events is much lower than that in summer and autumn, and hence the algal bloom samples in spring and winter are much less in number, leading to relatively poor identification results.

Strong generalization is an important capability that should be considered when developing deep learning-based methods. In the training of the proposed model, only the samples selected from the images from 2016 to 2019 were used. Therefore, it is necessary to validate whether the model is still robust and reliable for the images in other years; that is to say, it is necessary to validate whether the model has strong generalization in different years. In Table 4, the quantitative assessment results of the different methods for the spring and winter validation datasets and the summer and autumn validation datasets from 2020 are listed. Four remote sensing images from different seasons in 2020 and the corresponding interpretation maps using the different methods are displayed in Figure 5.

**Table 4.** Evaluation of algal bloom identification results in Chaohu Lake in 2020.

| Method | Spring and Winter | Summer and Autumn |
| --- | --- | --- |
| SVM | 75.77% | 78.42% |
| BPNN | 76.91% | 86.33% |
| SPCU_Net | 80.03% | 89.28% |
| DPCU-Net | 89.66% | 93.41% |

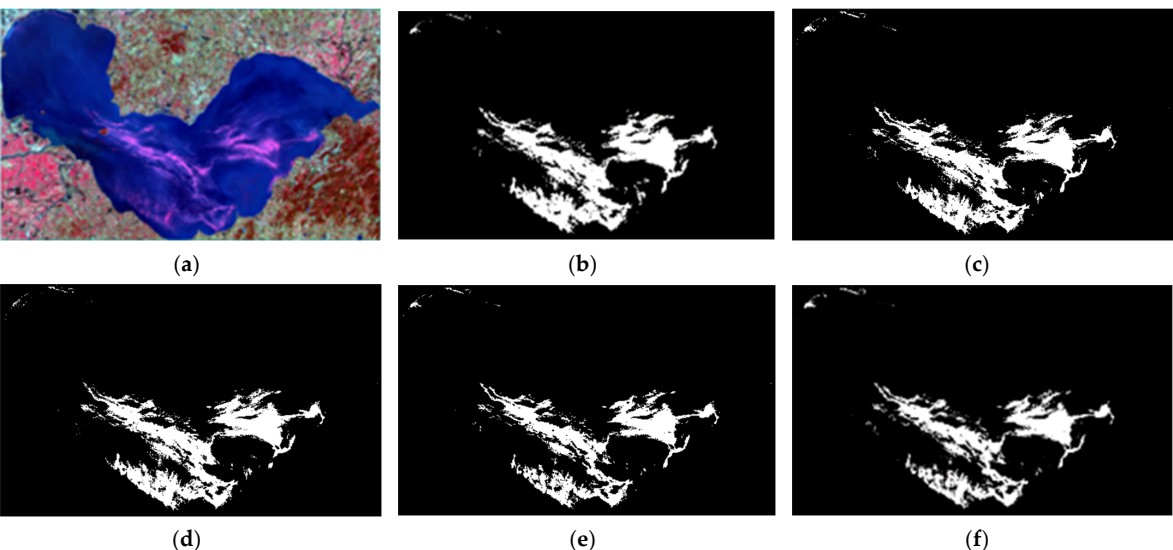

(a)　　　　　　　　(b)　　　　　　　　(c)

(d)　　　　　　　　(e)　　　　　　　　(f)

**Figure 5.** *Cont.*

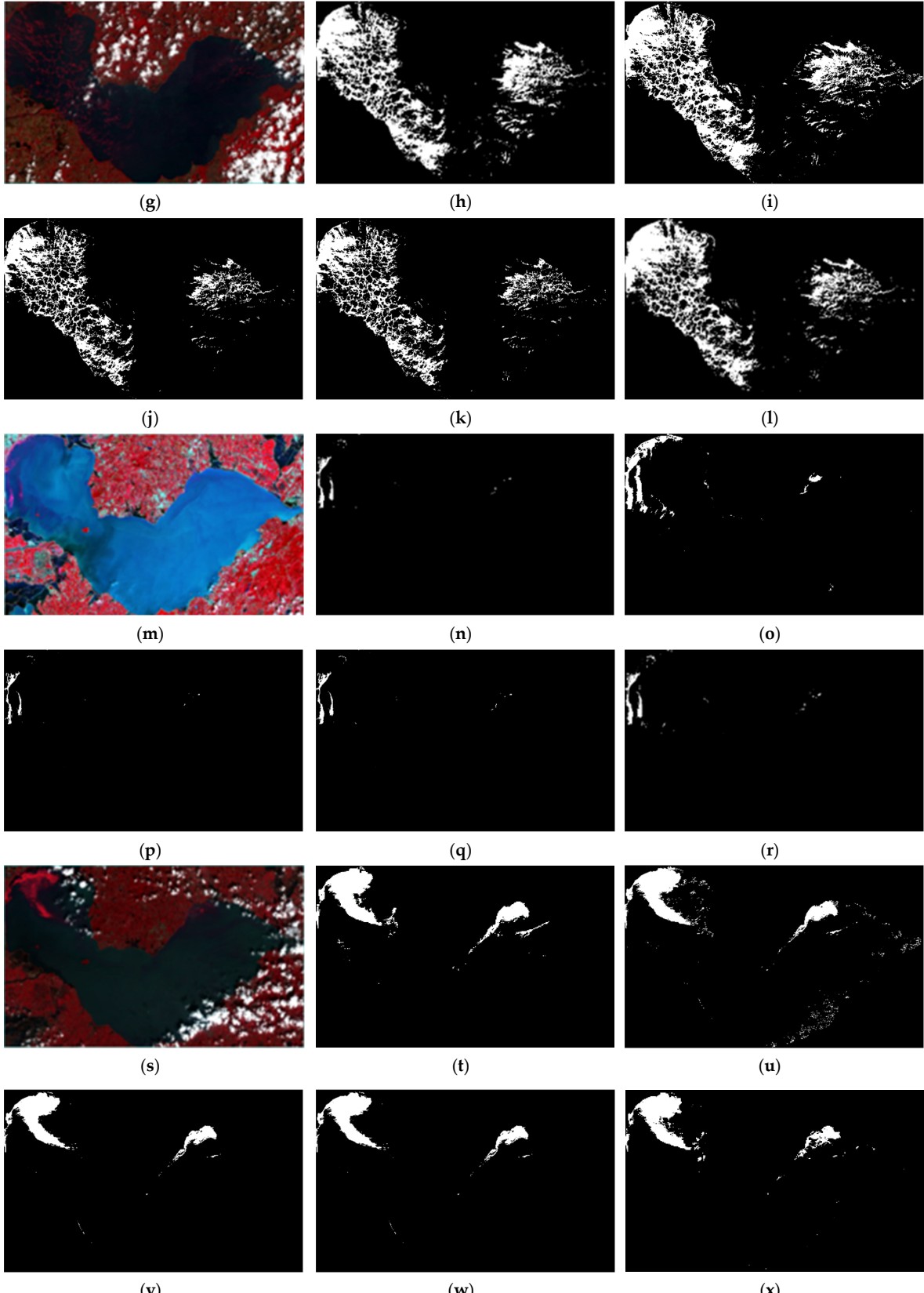

**Figure 5.** Algal bloom identification results for the images from 2020: (**a**) a remote sensing image of Chaohu Lake in spring; (**b**) a labeled image of (**a**); (**c–f**) the identification results using SVM, BPNN, SPCU_Net, and DPCU_Net, respectively; (**g**) a remote sensing image in summer; (**h**) a labeled image of (**g**); (**i–l**) the identification results using different methods; (**m**) a remote sensing image in autumn;

(**n**) a labeled image of (**m**); (**o**–**r**) the identification results using different methods; (**s**) a remote sensing image in winter; (**t**) a labeled image of (**s**); (**u**–**x**) the identification results using different methods.

Once again, from the quantitative assessment values, we can see that the proposed DPCU_Net performs best among all the methods, and the advantages of the deep learning-based methods with regard to the SVM method and BPNN method are still notable. Furthermore, as discussed before, the water bloom identification results for the summer and autumn dataset are better than that for the spring and winter dataset. By comparing the assessment values in Tables 3 and 4, we can easily notice that the performance of each method in processing the images from 2020 is slightly degraded. This is reasonable, since no training samples in these images were selected for all the methods. The water bloom identification maps in Figure 5 also demonstrate the superiority of the proposed method: the algal bloom identification map by the proposed method is highly in line with the real distribution state of algal blooms, and the profiles of the algal blooms are well retained.

### 4.3. Discussion

In the last two decades, some image classification methods, especially supervised machine learning methods, have been studied for the identification of algal blooms of lakes using remote sensing images. However, due to the use of insufficient training samples and the limited capability of extracting features from images, traditional supervised machine learning methods can hardly achieve high interpretative accuracy for algaes with complicated spectral reflection characteristics. Additionally, low computational efficiency is another drawback of the traditional methods.

The emerging and advanced deep learning theory has strong learning and generalization abilities, due to the use of sufficient training samples and refined networks. Deep learning-based methods introduce the possibility of people identifying algal blooms using remote sensing images rapidly and precisely. However, some limitations still exist when developing deep learning-based algal bloom monitoring methods. Firstly, although well-trained deep learning models need just a few seconds to interpret an image (that is to say, it is conveniently applied in image interpretation), the training process of the model is quite time consuming (normally requiring several hours) and has high requirements for the software and hardware configures of computers. Secondly, deep learning-based methods are big-data driven and their good performance highly depends on a sufficient number of samples. This means that, for lakes with small scales of water bloom outbreaks, the model developers can hardly obtain sufficient training samples, and in such a case, the performance of deep learning-based methods may even be poorer than traditional methods.

### 5. Conclusions

Rapid and accurate monitoring of algal blooms using remote sensing techniques is an effective means for the prevention and control of algal blooms. In this paper, taking Chaohu Lake as the study area, a dual U-Net deep learning model (including a U-Net network for spring and winter and a U-Net network for summer and autumn) is proposed for the identification of algal blooms, according to the traits of the algae in different seasons. First, the spectral reflection characteristics of the algae in Chaohu Lake in different seasons are analyzed, and sufficient samples from the images from 2016 to 2019 are selected for training the proposed model. Then, an attention gate architecture (including a position attention module and a channel attention module) is added to the classical U-Net framework, aiming at enhancing the capability of the network on feature extraction. Finally, the identification results are obtained by inputting remote sensing data into the model. The experimental results show that the proposed deep learning model achieves much better performance than the SVM and BPNN supervised classification methods, and also outperforms the single U-Net model using data from whole year as the training samples. Furthermore, the profiles of algal blooms are well-captured. In the experimental part, the images from 2020

are also used to validate the generalization capability of the proposed DPCAU-Net, and the results show that, even though the training samples of the model are selected from other years, the identification performance for the images from 2020 is still pleasing.

**Author Contributions:** Conceptualization, S.Z. and Y.W.; methodology, S.Z.; software, S.Z.; validation, S.Z.; formal analysis, S.Z.; investigation, S.Z.; resources, Y.W.; data curation, S.Z.; writing—original draft preparation, S.Z. and Y.W.; writing—review and editing, X.M.; visualization, S.Z.; supervision, S.Z.; project administration, S.Z.; funding acquisition, X.M. All authors have read and agreed to the published version of the manuscript.

**Funding:** This work was supported by the Hefei Municipal Natural Science Foundation No. 2021041.

**Institutional Review Board Statement:** Not applicable.

**Informed Consent Statement:** Not applicable.

**Data Availability Statement:** Not applicable.

**Conflicts of Interest:** The authors declare no conflict of interest.

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
