# Peer review of "Deep Learning-Based Algal Bloom Identification Method from Remote Sensing Images—Take China’s Chaohu Lake as an Example"

_sustainability, doi:10.3390/su15054545_

Round 1
Reviewer 1 Report
In this paper, taking Chaohu Lake as the study area, a dual U-Net model (including a U-Net network for spring and winter and a U-Net network for summer and autumn) is proposed for the identification of algal blooms from remote sensing images, according to the different traits of the algae in different seasons. Results seem fine but still there are many issue that can be address.
1) The rapid economic development and population growth have brought in some serious environmental problems, including the eutrophication of lakes which seriously endangers their ecological environment service function. - add a reference for this statement.
2) "Manual or automatic site sampling and monitoring method is a routine and intuitive method for water eutrophication monitoring. By the physical and chemical analyses of water samples, the traits of the algae can be directly and accurately obtained" - add a reference for this statement.
3) A number of methods have been presented in the last two decades to identify algal blooms in the lakes from remote sensing images- what are the names of the techniques?
4) "In the last few years, deep learning theory has attracted wide attentions in many fields. "- add the following reference for this such as (CXray-EffDet: Chest Disease Detection and Classification from X-ray Images Using the EfficientDet Model)
5) Clearly, deep learning theory provides the possibility for the rapid and accurate monitoring of algal blooms.- add justification for this statement.
6) What are the major contributions of this work? add under the introduction section.
7) "The attention mechanism in deep learning is similar to the human visual attention mechanism. By using minimal attention resources, the network can pay more attention to the main features of the targets, thus improving the efficiency and accuracy of feature information extraction."- add the reference for this.
8) What are the hyper parameters of this work? How many parameters of this work have been trained?
9) Add the analysis of this work under the results section.
Author Response
Please refer to the attched file.

Reviewer 2 Report
Dear authors, I read your manuscript with pleasure. I think there is no need to change or supplement it. I suggest publishing it as is.
Best regards.
Author Response
Dear reviewer:
We sincerely thank you for your approval of our work. The other two reviewers have put forward some constructive and helpful suggestions, and we have made the changes as the reviewers suggested, which further improved the quality of our paper.
Best regards.
Reviewer 3 Report
The current manuscript entitled “Deep Learning-Based Algal Blooms Identification Method from Remote Sensing Images—Take Chaohu Lake as an Example” by Zhu et al. explored the application of deep learning based-technique for the algal blue identification in Take Chaohu lake, China. After a careful reading, I found this work interesting and suitable for publication in the Sustainability journal. The paper is well-written and presents a detailed description of the research problem. The authors provide an overview of the existing research and provide background information on Chaohu Lake. They then discuss their proposed method, which involves using a convolutional neural network and a tiling approach to detect algal blooms. The authors provide a step-by-step description of the experiment, which is clearly explained and easy to follow. The results of the experiment are also presented in a straightforward manner. The paper is technically sound and the authors provide adequate justification for their approach. However, the proposed method of algal bloom detection is less compared to other existing techniques, providing a less-thorough evaluation of the results in a proper discussion section. Considering these facts, I suggest acceptance of this manuscript after minor revisions. My specific comments are:
1. Add the country name “China” in the title.
2. The abstract should be modified providing major numerical findings and their relevance to the proposed method. The problem statements can be reduced and more focus on your major results should be given.
3. Major sources and elements which contribute to algal blooms should be given in the introduction section. It also provides the role of algae in aquatic environments as key organisms as they also contribute to ecological restoration and feed several other trophic levels.
4. Define “/” in tables.
5. Fig quality is commendable.
6. I have a major concern regarding the discussion of obtained results. I suggest modifying “4. Results” to “4. Results and discussion” and comparing your findings with recent reports.
7. The conclusion can be reduced to 250 words and major texts can be shifted to the discussion.
Round 2
Reviewer 1 Report
Authors well addressed my comments and I recommend to publish this work now in this well reputed journal. thank you